# Synthesis of Ultrastable Gold Nanoparticles as a New Drug Delivery System

**DOI:** 10.3390/molecules24162929

**Published:** 2019-08-13

**Authors:** Florence Masse, Pascale Desjardins, Mathieu Ouellette, Camille Couture, Mahmoud Mohamed Omar, Vincent Pernet, Sylvain Guérin, Elodie Boisselier

**Affiliations:** Research Center of CHU de Québec, CUO-Recherche and Université Laval, Faculty of Medicine, Ophthalmology Department, chemin Ste-Foy 1050, G1S 4L8, Québec (Québec), Canada

**Keywords:** ultrastability, gold nanoparticles, drug delivery, MTS assay, wound healing assay

## Abstract

Nanotechnologies are increasingly being developed for medical purposes. However, these nanomaterials require ultrastability for better control of their pharmacokinetics. The present study describes three types of ultrastable gold nanoparticles stabilized by thiolated polyethylene glycol groups remaining intact when subjected to some of the harshest conditions described thus far in the literature, such as autoclave sterilization, heat and freeze-drying cycles, salts exposure, and ultracentrifugation. Their stability is characterized by transmission electron microscopy, UV-visible spectroscopy, and dynamic light scattering. For comparison purposes, two conventional nanoparticle types were used to assess their colloidal stability under all conditions. The ability of ultrastable gold nanoparticles to encapsulate bimatoprost, a drug for glaucoma treatment, is demonstrated. MTS assays on human corneal epithelial cells is assessed without changing cell viability. The impact of ultrastable gold nanoparticles on wound healing dynamics is assessed on tissue engineered corneas. These results highlight the potential of ultrastable gold nanoparticles as a drug delivery system in ocular therapy.

## 1. Introduction

Gold nanoparticles (GNPs) are known as highly tunable materials and are studied in various research fields such as nanomedicine [1,2,3,4,5,6,7,8,9]. However, in many instances, GNP instability restricts their use in this specific field. As an example, the weak stability against physiological buffering salts excludes the use of many GNP types for biomedical applications, since blood is a highly ionic media [10]. In recent decades, intensive research has been undertaken to develop new synthesis leading to ultrastable GNP. An exhaustive review of highly-stable and ultrastable GNPs led to only a dozen publications describing new synthesis conditions and resistance against different criteria [11,12,13,14,15,16,17,18,19,20,21,22]. However, most of the time, GNPs presented ultrastable properties only under limited conditions and were not water-soluble. Indeed, there is no precise definition of what should be called ultrastable, and this term can be misused. For instance, numerous GNPs have been reported to be stable for more than six months at room temperature without any alteration [18,23], but this feature should not be sufficient to define them as ultrastable [12,16,20]. The notion of time must be taken into account and the stability kinetics explained when the aggregation is observed after a few minutes following salt exposure [11,13,16,17,19,20]. Nonetheless, some studies have suggested a degree of stability against freeze drying, salts resistance [11,13,15,16,17] and/or autoclaving [24]. A broader overview of GNP characteristics is preferable for medical applications. Moreover, none of the aforementioned GNPs maintained their stable properties after sterilization by autoclaving in an aqueous buffer. In this study, we propose defining the term “ultrastable” in this context as being GNPs whose characteristics (plasmon band, hydrodynamic diameter, and core diameter) are not modified under drastic conditions, such as freeze drying, heating, ultracentrifugation, solubilization in ionic media, as well as autoclave sterilization.

Among the many synthesis methods leading to gold salt reductions to spherical GNPs, the procedures described by Brust and Turkevish remain the most widely used [25,26]. The Brust method is a single step synthesis using sodium borohydride (NaBH_4_) as a reducing agent. It usually leads to smaller nanoparticles than those produced by the Turkevich method, which uses citrate as both the reducing and stabilizing agent. Although GNP synthesis could be performed directly in water, this was found to form few stable nanoparticles, and only to work with thiolated, low molecular weight ligands that could quickly stabilize the gold core. In order to improve synthesis efficiency, GNPs are often synthesized in organic solvents, and then transferred into water prior to the purification steps. Indeed, solvent molecules can stabilize the gold core acting as a small temporary ligand.

In this study, we report a new synthesis method for ultrastable water-soluble GNPs that can support harsh conditions such as several cycles of freeze drying, heating, ultracentrifugation, solubilization in ionic media, as well as autoclave sterilization. For comparison purposes, two other GNP types (thiolated PolyEthylene Glycol (PEG) or citrate coated GNP), were synthesized and subjected to the same harsh treatments. Their stability was assessed using different methodologies, including UV-visible spectroscopy, Dynamic Light Scattering (DLS), and Transmission Electron microscopy (TEM). As a biomedical application, we report the encapsulation of bimatoprost, a prostaglandin analog commonly used in ophthalmology to control intraocular pressure in glaucoma, reflecting the potential of GNP as a new drug delivery system for ophthalmic therapy. The encapsulated drug was quantified using a new protocol with magnetic beads, and the cytotoxicity of the nanoparticles was evaluated using a MTS assay. Finally, a wound-healing assay was performed on human tissue engineered corneas to determine the impact of ultrastable GNPs on the wound-healing dynamic.

## 2. Results

### 2.1. Synthesis

The new synthesis conditions involve the use of acetonitrile in a one-step procedure without the use of any phase transfer/stabilizing agent. The use of thiol groups on the gold core leads to enhanced stability, and should thus be used for in vivo applications [25,27,28]. The chosen ligands for GNP stabilization are thiolated PEG groups, generally biocompatible, and leading to water-soluble materials [29]. Three PEG groups, having different molecular weights (800, 2000 and 6000 g.mol^−1^), have been used to functionalize GNP 800, GNP 2000 and GNP 6000, respectively. The initial mixing of chloroauric acid (HAuCl_4_) and PEG results in a partial reduction of gold (Scheme 1 and part 3.2. of Materials and Methods section). The complete reduction is achieved upon addition of NaBH_4_, that initiates crystal formation. The gold atoms are then capped and stabilized with thiolated ligands. The speed of crystal growth versus ligand capping determines the success of this synthesis. If the speed of crystal growth is faster than the speed of capping, the particles will not be properly stabilized and will precipitate. The presence of acetonitrile in the reaction mixture slows crystal growth due to nitrogen atoms acting as partial stabilizers. The lower reducing strength of NaBH_4_ in this solvent compared with water also contributes to slow crystal growth, leading to a better capping, and making the GNPs more stable [30].

UV-visible spectra reveal GNP surface plasmon resonance absorbance peaks centered at 515 nm for GNP 800 and 514 nm for both GNP 2000 and GNP 6000 (Figure 1 and part 3.5 from Materials and Methods section). The molar extinction coefficients of the GNP were calculated according to the Lambert-Beer law by measuring their maximal absorbance at different concentrations (Appendix A). The molar extinction coefficients of GNP 800, GNP 2000 and GNP 6000 are respectively 618 204 L.mol^−1^.cm^−1^, 545 219 L.mol^−1^.cm^−1^, and 3 711 567 L.mol^−1^.cm^−1^ (Table 1). Their gold core, observed by TEM, has a diameter of 3.1 ± 1.7 nm, 4.9 ± 1.7 nm, and 4.0 ± 1.8 nm, respectively (Figure 1, Table 1 and part 3.6 from Materials and Methods section). DLS measurements report hydrodynamic diameters of 23 ± 4 nm, 35 ± 2 nm, and 53 ± 2 nm for GNP 800, GNP 2000, and GNP 6000, respectively (Figure 1, Table 1 and part 3.7 from Materials and Methods section). Thus, the polymer layer has a radius of about 20 nm, 30 nm, and 49 nm for GNP 800, GNP 2000, and GNP 6000, respectively. The three types of GNPs have a neutral charge, largely due to the neutrality of PEG. Indeed, the zeta potential of GNP 800, GNP 2000, and GNP 6000 are −0.1 ± 1.0 mV, −0.2 ± 0.5 mV, and −1.8 ± 4.3 mV, respectively (Table 1 and part 3.7 from Materials and Methods section). Elemental analysis by ICP-OES revealed different weight percentages of gold and sulfur for each type of gold nanoparticles. GNP 800, GNP 2000 and GNP 6000 were found to contain 73.0%, 54.1%, and 49.6% of gold atoms and 1.5%, 0.5%, and 1.4% of sulfur atoms, respectively (Table 1 and part 3.8 from Materials and Methods section). Combining these data with the diameter of the gold core, molecular weights of 146 922 g.mol^−1^, 616 059 g.mol^−1^, and 1 805630 g.mol^−1^ were deduced for GNP 800, GNP 2000, and GNP 6000, respectively (Table 1 and part 3.2 from Materials and Methods section).

One can notice that the core diameter, as well as the number of ligands, is maximal for GNP 2000, whereas the % S atoms is minimal. This observation could be explained by the difference of behavior of GNP according to their molecular weights, in particular during the stabilization of the gold core. While PEG alone in water can have Gaussian coil or flat plate configurations [31], they can stabilize molecules or proteins with wrapped or expanded configurations [32]. During GNP growth (Scheme 1), the three types of PEGs (800, 2000 and 6000) may behave differently, leading to core diameters ranging from 3.1 ± 1.7 to 4.9 ± 1.7 nm, which are nevertheless quite similar. PEG 6000 are more likely to undergo steric hindrance, and the ability of the thiol group to reach the gold core could thus be diminished. Steric hindrance could also lead to major stabilization, thus producing smaller cores than expected. Furthermore, as PEGs can reduce and stabilize GNPs by themselves [33], large ones, such as PEG 6000, could lead to smaller gold cores due to the premature reduction of gold. As a consequence of the smaller gold core of GNP 6000, the number of stabilizing ligands around the nanoparticles is also reduced, giving GNP 2000 the largest gold core diameter and number of ligands observed in this study. With this largest core diameter, GNP 2000 contains 3 621 gold atoms, which is much more than the two other types (918 and 1970 respectively for GNP 800 and GNP 6000). Despite a number of sulfur atoms, i.e., 165, this only represents 0.5% (in mass) in this case.

Two other types of GNPs were synthesized using the modified Brust method, as described by Ouellette et al. [8], and the Turkevich method, following protocols described by Turkevich et al. [26] (see parts 3.3 and 3.4 from Materials and Methods section). The first type corresponds to GNPs stabilized with thiolated PEG 2000 groups, for which the synthesis is performed in methanol and water, and therefore, not stabilized with acetonitrile during the crystal growth. This synthesis yielded a 6.7 ± 2.8 nm gold core, with a hydrodynamic diameter of 14.3 ± 0.3 nm. They are referred to as Non UltraStable (NUS) GNPs. The second type of GNP is stabilized with citrate ligands, and is referred to as GNP CIT. As previously stated, the citrate ligands only weakly stabilize the gold core during synthesis [26]. Core and hydrodynamic diameters of respectively 4.1 ± 1.9 nm and 29.1 ± 0.2 nm were obtained. Elemental analysis, zeta potential and the determination of the number of ligands, molecular weights and molar extinction coefficients were previously described in the literature ([8] for NUS GNP and [26,34,35,36] for CIT GNP).

### 2.2. Ultrastability Assays

The GNP plasmon band is very sensitive to any change occurring around the GNP (core diameter, ligands number, pH, temperature, aggregation, etc.) and would reveal any perturbation by a wavelength shift or an intensity change [2,37]. GNP stability results are shown on Figure 1 for all GNP types.

UV-visible spectra were obtained with 0.044 mg.mL^−1^ of GNP before and after three cycles of 24 h of freeze drying, three periods of 12 h of heating at 65 °C, three precipitations by ultracentrifugation, sterilization by autoclave, and when dissolved in PBS during more than one week (Figure 1a–e, Appendix A and parts 3.10 to 3.14 from the Materials and Methods section). Figure 1a–c shows the stability results for GNP 800, 2000, and 6000, respectively (see Appendix A for the UV-visible spectrum and the TEM image of GNP 800 in full size). The UV-visible spectrum for each GNP observed after treatment, represented by dotted grey lines, is almost the same as the initial one, represented by black full lines (see Appendix A for wavelength and absorbance values of plasmon band maximum of all UV-visible spectra).

While very small variations can be observed for GNP 800 and GNP 2000 after heating, freeze drying or autoclaving, GNP 6000 is very resistant to any condition. We hypothesize that longer PEG groups reduce the access to the core of the nanoparticles and better stabilize GNP than shorter PEG groups. The size distribution analysis, extracted from TEM images using the ImageJ software, allowed us to determine the core size area, and thus the core diameter (Figure 1). The mean core of ultrastable GNP diameter does not significantly vary after each treatment. DLS experiments led to quite similar results after all treatments, and presented no signs of aggregation. The plasmon bands observed in UV-visible spectroscopy also remain similar (see Appendix A for wavelength and absorbance values). Indeed, variations of ±1 nm for the wavelength and ±0.02 a.u. for the absorbance are observed, that are inferior to the resolution of the spectrophotometer. Only autoclave sterilization leads to a slight shift of the wavelength to 520 nm for the three types, but no drastic change was observed for core and hydrodynamic diameters, as was the case for GNP NUS and GNP CIT. For these latter, UV-visible spectra, TEM images and DLS measurements were obtained according to the same methodology. Results are presented in Figure 1d,e, respectively. For GNP NUS, the UV-visible spectrum is not affected by the ultracentrifugation treatment. This is confirmed by TEM experiments as the core diameter remains similar (6.7 ± 2.8 nm before and 5.5 ± 2.4 nm after ultracentrifugation). However, the hydrodynamic diameter seems amplified after treatment, because of their low stability towards this treatment. While the UV-visible spectra remain stable after the heating treatment, both TEM and DLS revealed a significant reduction of the GNP core and hydrodynamic diameters, suggesting a reorganization of these GNP during the treatment. GNP NUS instability is confirmed by the UV-visible spectra, TEM images and DLS after freeze drying, autoclaving and salts treatments, where an increase of 7% and a decrease of 11% and 20% are respectively observed for the absorbance values of their plasmon bands (see Appendix A). As the GNP NUS plasmon band is still observable in each case, we conclude that such treatments do not lead to their aggregation but to a reorganization into GNPs with different properties. Furthermore, GNP CIT were found to be unstable after every treatment (Figure 1e). In most cases, their plasmon band was highly modified or missing. Indeed, the absence of the plasmon band occurs when GNPs aggregate and no longer absorb light at around 515 nm, rendering further analysis such as TEM and DLS impossible. While the plasmon band of GNP CIT does not seem to be affected by the heating treatment, DLS and TEM analyses could not be conducted due to very polydispersed GNP size distribution caused by GNP aggregation. By comparison, GNP NUS and GNP CIT do not remain intact after the same treatments as those applied to our new ultrastable GNP.

To our knowledge, the ultrastable GNP 800, 2000, and 6000 are the first reported GNPs that can sustain autoclave sterilization without any major change in their UV-visible spectra, and both core and hydrodynamic diameters. This property is very promising for their use in nanomedicine. Indeed, sterilization techniques based on heat, irradiation, or chemical sterilization do not provide the same efficiency of sterilization. Autoclave is one of the most powerful techniques to obtain sterile materials, along with plasma and ethylene oxide treatments [38]. Indeed, autoclave makes possible the deactivation of mycobacteria, vegetative bacteria, bacterial spores, non-enveloped and enveloped viruses, prions, and fungi. Finally, the stability of GNPs against PBS is another essential property supporting their potential biological use, since this saline buffer mimics highly-ionic media such as those found physiologically. These ultrastable GNPs could thus be appropriate for biomedical applications, such as drug delivery systems.

### 2.3. Drug Encapsulation and Quantification of the Encapsulated Drug

Nanomedicine currently needs ultrastable nanomaterials that can remain intact under the different conditions applied in drug formulation and delivery. GNPs have a promising future in biomedical applications in ophthalmology [9]. The encapsulation of an ophthalmic drug by GNP was therefore tested to assess their potential use as nanovectors in this context. When medication is administered onto the cornea, very few molecules, that is 0.0006 to 0.02%, reach the anterior chamber of the eye, which limits drug efficiency [8]. The mucoadhesive properties of GNP could be used to encapsulate active molecules and maintain them at the corneal surface to allow the slow release of the drug to occur and the improvement of their efficacy [8]. As an application, the ability of GNPs to encapsulate drugs was evaluated with bimatoprost, a pharmacological drug commonly used to treat glaucoma. Bimatoprost is administered through ocular drops at high doses, and the use of a drug delivery system could improve this medication by decreasing the required concentration of the active agent in the formulation. Indeed, a high drug concentration may be irritating for the cornea. Moreover, a majority of these molecules are hydrophobic, and therefore, necessitate the use of additives in their formulation to allow their biodisponibility. The encapsulation of active molecules in GNP should thus improve their solubility.

The pure drug was first mixed with GNP, as detailed in part 3.15 in from Materials and Methods section. The separation of the drug-loaded GNP from the free drug is required to determine the encapsulation efficiency. As mentioned before, the gold core of GNP is affected by other molecules in solution, leading to variations in the absorbance intensity that render quantification very difficult. The removal of the nanoparticles is commonly done by pelleting them by centrifugation. However, in the present case, the GNPs are very stable in solution, and the force required to pellet them (356 000 g) may also lead to drug release. To overcome this issue, an immunoprecipitation protocol using magnetic beads (coated with protein A and linked to an Anti-PEG antibody) was designed to isolate the ultrastable GNPs (Figure 2a and part 3.15 from Materials and Methods section). To our knowledge, this is the first time that magnetic beads have been used for GNP purification. The immunoprecipitation was first performed on GNP alone to validate their complete removal, and on bimatoprost alone to assess any unspecific interaction with the antibody (the complete protocol for immunoprecipitation of the GNP can be found in part 3.15. of Materials and Methods section). The following experiments were done on GNP 2000 exclusively. Briefly, GNPs were mixed with bimatoprost with GNP/drug ratios of 1/126, 1/252, and 1/627 for 72 h. Conditioned magnetic beads were added to the solution for 24 h, and the GNPs were then separated using four magnetizations (Appendix A). The free drug was quantified by HPLC and compared with a calibration curve (Appendix A). The encapsulation efficiency of the GNPs is detailed in Figure 2b. GNP/drug ratios of 1/126, 1/252, and 1/1627 lead respectively to encapsulation efficiencies of 67 ± 6%, 51 ± 8% and 33 ± 7%, corresponding to a number of encapsulated drug molecules per nanoparticle of 84 ± 8, 129 ± 20s and 207 ± 44. Increasing the initial number of drug molecules mixed with a given number of GNPs thus leads to a decrease of the encapsulation efficiency, but to a larger number of encapsulated drug molecules per nanoparticle. These data confirm the ability of ultrastable GNPs to encapsulate drugs and their potential to be used as a drug delivery system.

### 2.4. Toxicity and Wound Healing Assays

In order to determine the range of concentrations for which ultrastable GNPs are safe for the cells, a MTS assay was performed according to a protocol previously published [39]. The cell viability obtained for each concentration of GNP, according to the GNP type, is detailed in Figure 3. Each band corresponds to the average value obtained for three different populations of hCECs. The data suggest that none of the three types of GNP significantly reduces cell viability upon exposure for 18 h, at all tested doses.

The inertness of GNPs on biological mechanisms was investigated through the study of the influence of ultrastable GNPs on the wound healing properties of hTECs [39]. To that purpose, hTECs were wounded using an 8 mm trepan and left to recover in complete culture medium in either the absence or presence of GNPs. The dynamic of wound closure is shown on Figure 4a,b. No significant difference was observed in the time required for complete wound closure between control (solid line) and GNP 2000-treated (50 µL droplet of 88 nM GNP 2000 per day) hTECs (dotted line). Furthermore, this dose of ultrastable GNPs had no apparent impact on this hTECs biological mechanism, which is promising for their future use in ophthalmic treatments.

## 3. Materials and Methods

### 3.1. Chemicals

Gold chloride trihydrate (HAuCl_4_∙3H_2_O), sodium borohydride (NaBH_4_), chlorhydric acid (HCl), nitric acid (HNO_3_), sodium chloride (NaCl), potassium chloride (KCl), sodium phosphate dibasic (Na_2_HPO_4_), potassium phosphate monobasic (KH_2_PO_4_), isopropanol, and acetonitrile were all purchased from VWR International (Ville Mont-Royal, QC, Canada). Polyethylene glycol methyl ether thiol with a molecular weight of 2000 g.mol^−1^ (referred to as PEG 2000) was purchased from Laysan Bio (Arab, AL, USA). Polyethylene glycol methyl ether thiol with molecular weights of 800 and 6000 g.mol^−1^ (referred to as PEG 800 and 6000, respectively) and ascorbic acid were purchased from Sigma-Aldrich (St. Louis, MO, USA). Anti-Polyethylene glycol antibody [PEG-B-47] was purchased from Abcam (Toronto, ON, Canada). Surebeads magnetic beads with protein A were bought from Biorad Laboratories (Mississauga, ON, Canada). Tween-20 and potassium nitrate (KNO_3_) were purchased from Fisher Scientific (Ottawa, ON, Canada). Bimatoprost was bought from Cedarlane Laboratories (Burlington, ON, Canada). All the glassware used to synthesize GNPs was thoroughly washed with aqua regia (3:1 HCl:HNO_3_) and rinsed with nanopure water prior to experimentation. 3-(4,5-dimethylthiazol-2-yl)-5-(3-carboxymethoxyphenyl)-2-(4-sulfophenyl)-2*H*-tetrazolium (MTS) was obtained from Promega (Madison, WI, USA). Normal eyes were obtained from the Banque d’Yeux Nationale of the Centre Universitaire d’Ophtalmologie; CHU de Québec, Hôpital du Saint-Sacrement, Québec, QC, Canada. Murine Swiss-3T3 fibroblasts were obtained from ATCC (Rockville, MD, USA).

### 3.2. Ultrastable Gold Nanoparticles Synthesis

Stirring at 400 rpm, 250 µL of an aqueous solution of 0.1 mg.mL^−1^ of gold chloride trihydrate (0.06 mmol) was added to 15 mL of a 1:1 acetonitrile:isopropanol mix. 0.01 mmol of PEG was added to 30 mL of isopropanol prior to addition to the gold solution. The resulting solution was mixed for an hour. A fresh solution of 0.028 g of NaBH_4_ (0.74 mmol) in 10 mL of ice-cold water was added dropwise to the gold solution using a peristaltic pump (1 mL/min), while stirring at 700 rpm. Then, the mixture was stirred at 400 rpm for three hours to allow the nanoparticles to grow and stabilize. The solution was evaporated with a rotary evaporator under reduced pressure. Next, 10 mL of water was added to re-suspend the nanoparticles. Their purification was performed by dialysis (molecular weight cut-off of 12,000 g·mol^−1^) for 3 days, changing the water outside the membrane at least 4 times. To determine the concentration of the gold nanoparticles in the resultant solution, a known volume of the solution was freeze dried and weighted (n = 3).

### 3.3. GNP NUS Synthesis

GNP NUS coated with PEG 2000 were synthesized using a modified Brust method [8]. Briefly, 0.127 mmol of a aqueous gold chloride solution of (HAuCl_4_∙3H_2_O) was added to 30 mL of a methanol:water (1:1) solution. Then, 0.024 mmol of PEG 2000 was added, and the solution was agitated for 90 min at 400 rpm. Next, 0.040 g (1.057 mmol) of NaBH_4_ dissolved in 20 mL of ice-cold water was added dropwise at a rate of 1 mL.min^−1^ under 700 rpm stirring using a peristaltic pump. The mixture was stirred for 3 h at 400 rpm. A rotary evaporator was used to evaporate the methanol under reduced pressure. Using a minimal amount of a saturated NaCl solution, the nanoparticles were extracted with dichloromethane. GNPs were then dissolved in 10 mL of water and dialyzed over two days.

### 3.4. GNP CIT Synthesis

GNP CIT were synthesized according to Turkevich et al. [26]. Briefly, 1% chlorauric acid aqueous solution was brought to boil prior to the addition of a 1% solution of sodium citrate under 400 rpm agitation.

### 3.5. UV-Visible Spectroscopy

UV-visible spectra were collected from 200 to 800 nm using a Cary Eclipse 50 Bio UV-vis spectrophotometer from Varian (Winnipeg, MB, Canada). A quartz cuvette (1 mm × 10 mm pathlength) from Hellma (#104.002-QS) was used (Markham, ON, Canada). The molar extinction coefficients of GNP were calculated according to the Lambert-Beer law by measuring the absorbance of GNP at 515 nm for different concentrations (Appendix A).

### 3.6. Transmission Electron Microscopy (TEM)

Copper grids covered with a vaporized carbon film were purchased from Ted Pella (California, USA). GNPs were diluted to a concentration of about 0.01 mg.mL^−1^ and a 5 µL droplet was deposited on each grid and left for a minute before removing the excess with a filter paper. The image acquisition was performed on a JEM 1230 from JEOL microscope (Tokyo, Japan). The voltage was set to 80 kV and a 50 000× zoom factor was used. To characterize the gold core size, at least 150 GNP were counted and analyzed for each sample, using ImageJ software.

### 3.7. Dynamic Light Scattering (DLS) and Zeta Potential Measurements

DLS and Zeta potential measurements were performed with the NanoBrook Omni from Brookhaven Instruments Corporation (Holtsville, NY, USA). For DLS measurements, each GNP types were diluted to concentrations of 0.044 mg/mL and 0.029 mg.mL^−1^ in 10 mM KNO_3_. Samples were filtered through a 0.2 μm pore size filter prior to analysis. The apparatus was set to an angle of 90° at 25 °C. After an equilibrium time of 10 min, ten measurements of 120 s were performed for each sample and the values were reported as Effective Diameter. Eppendorf disposable plastic cuvettes (#952010051) were used for analysis (Mississauga, ON, Canada). The treatment of the size distribution was performed using the CONTIN algorithm. GNP solutions diluted to concentrations of 0.029, 0.015, 0.007, and 0.004 mg.mL^−1^ allowed Zeta potential data to be collected. BI-SCP disposable plastic cuvettes and the BI-ZEL electrode assembly from Brookhaven Instruments Corporation were used (Holtsville, NY, Canada). The wavelength was set to 640 nm and the temperature was maintained at 25 °C. Run periods lasted 15 cycles, for 10 measurements, with an equilibration time of 10 min. The Smoluchowski model was used for analysis.

### 3.8. Elemental Analysis

Elemental analysis was conducted with Inductively Coupled Plasma Optical Emission Spectrometry (ICP-OES), requiring the oxidation of gold prior analysis. To do so, GNP were diluted to a 1:20 aqua regia:water ratio and a final GNP concentration of 0.85 mg·mL^−1^. Blank standards also containing 5% aqua regia were analyzed to confirm the absence of contamination specific to the method. The sample preparation was performed in 15 mL tubes from Sarstedt (#62.554.100) (Montreal, QC, Canada). They were heated to 90 °C, with care being taken to avoid boiling. The tube caps were left on the tubes, closed by a quarter of a turn. The apparatus used for the experiments was the ICP-OES-5110, from Agilent, in Radial mode (Mississauga, ON, Canada). Calibration methods of yttrium internal standard and standard addition were done to prevent matrix effects. Acquisition was achieved at 242.8 nm and 181.9 nm for gold and sulfur quantification, respectively.

### 3.9. Number of Ligand and Molecular Mass Determination

The molecular weights of GNP were estimated as previously published [40]. Assuming monodisperse spherical gold nanoparticles, the volume (V) of the metallic core can be estimated according to:(1)$$V=\frac{4\pi \times {\left(\mathrm{radius}\;\mathrm{in}\;\text{Å}\right)}^{3}}{3},$$
The number of gold atoms present in the gold core, n (Au), can be estimated according to:(2)$$n\;\left(\mathrm{Au}\right)=\frac{V\;\mathrm{in}\;{\text{Å}}^{3}}{17{\;\text{Å}}^{3}}$$
With the molecular weight (MW) of Au and S, the elemental analysis allows the estimation of the number of thiol groups, n (S), on the surface according to:(3)$$\frac{n\;\left(\mathrm{Au}\right)\times \;\mathrm{MW}\;\left(\mathrm{Au}\right)}{\;\%\;\mathrm{Au}\;\mathrm{atoms}}=\frac{n\;\left(S\right)\times \mathrm{MW}\;\left(S\right)}{\;\%\;S\;\mathrm{atoms}}$$
Finally, the molecular weight of GNP can be estimated according to:(4)MW (GNP)=(n (Au)×MW (Au))+(n (S)×MW (ligand))

### 3.10. Stability against Freeze Drying

Two milliliters of GNPs were frozen at −80 °C for 5 h. The samples were then freeze dried for 24 h per cycle, for 3 cycles. 2 mL of water was then added to dissolve the dry GNP before analysis.

### 3.11. Stability against Heat

GNPs were heated in an incubator at 65 °C overnight. The sample was homogenized by shaking the tube upside down to collect the drops of water in the cap, before being exposed to two other heating cycles.

### 3.12. Stability against Ultracentrifugation

GNP were centrifuged for 45 min at 356,000 *g*. Centrifugations were performed on an Optima Max ultracentrifuge from Beckman Coulter Inc, using a TLA-120.2 rotor with polycarbonate centrifuge tubes (#343778) (Indianapolis, IN, USA). After centrifugation, the sample was remixed before facing two other cycles of ultracentrifugation.

### 3.13. Stability against Autoclave Sterilization

GNP were autoclaved at 121 °C for 15 min, with a total run time of 52 min. To counter the evaporation observed during the process, the remaining volume was precisely measured and adjusted to the initial value.

### 3.14. Stability against Phosphate-Buffered Saline (PBS)

GNPs were dissolved in PBS 1X. UV-visible spectra were recorded (Appendix A) after 0 h, 0.5 h, 1 h, 24 h, 48 h, 72 h, 96 h, 144 h, and 168 h in amber glass vials from Supelco (#29654-U) (Sigma, Oakville, ON, Canada).

### 3.15. Drug Encapsulation Protocol

#### Determination of the Encapsulation Time

Bimatoprost and GNP 2000 were mixed with a ratio of 1/126 during 72h and UV-visible spectra were then collected. Spectra of bimatoprost alone and GNPs alone were also recorded as controls to confirm that the changes were all due to the perturbation of the gold core.

#### Quantification of the Encapsulated Drug

The resultant solution was analyzed by High Performance Liquid Chromatography (HPLC) and compared to a calibration curve of bimatoprost (Appendix A). Encapsulation efficiency was determined indirectly. Drug concentrations of 8.6, 17.1 and 42.9 µM were used for GNP/drug ratios of 1/126, 1/252 and 1/627, respectively.

#### Immunoprecipitaton of the Nanoparticles (Adapted from SureBeads Protocol)

A. Washing of the beads (to remove sodium azide and tert-butyldimethylsilyl ethers used as preservatives)

Add 100 µL of magnetic beads to a tube, magnetize and remove the liquid. Add 1 mL of PBS (1X) with 0.0005% of Tween-20 (PBS-T) to the beads and homogenize by vortexing the sample for 30 sec. Magnetize and discard the liquid. Repeat this step 2 more times.

B. Addition of the antibody on the magnetic beads

Add 8 μg of antibody in 300 μL of PBS-T and suspend the beads. Shake for 10 min at room temperature with a Thermomixer from Eppendorf (Germany) at 1000 rpm. Magnetize the beads and remove the liquid. Wash twice.

C. Addition of the sample containing the GNP

Add the 0.5 mL sample containing GNP. Shake for 24 h at room temperature with a Thermomixer at 1000 rpm. Magnetize the beads and collect the supernatant. Put it in another tube, and then re-magnetize to remove all the nanoparticles (see Appendix A). Four magnetizations were used for the determination of the encapsulation efficiency.

### 3.16. MTS Assay

MTS assays were conducted as previously reported [39]. Briefly, Human Corneal Epithelial Cells (hCECs) (1 × 10^4^) were seeded with irradiated murine 3T3 fibroblasts (2 × 10^3^) in 96-wells plates in DH supplemented medium and incubated at 37 °C for 4 h prior to the addition of GNPs. The doses were chosen arbitrarily, since no cytotoxic data were available for the new ultrastable GNPs. For GNP 800, we used concentrations from 0.5 to 406 nM of GNP 800, corresponding to doses ranging from 1 × 10^7^ to 1 × 10^10^ GNP/cell. For GNP 2000, concentrations were from 0.1 to 93 nM of GNPs, corresponding to 3 × 10^6^ to 2 × 10^10^ GNP/cell. Then for GNP 6000, we tested concentrations from 0.3 to 279 nM of GNPs, corresponding to 7 × 10^6^ to 7 × 10^10^ GNP/cell. Cells were exposed to GNPs for 18h prior to the removal of the medium. To evaluate the cell viability, the cells were then exposed to MTS to complete the colorimetric assay. MTS (2 mg mL^−1^) was added to each well and the plates incubated at 37 °C for 4 h. Optical density (OD) was then measured using a microplate reader (Biorad Model 550, Mississauga, ON, Canada) at a wavelength of 490-nm.

### 3.17. Wound Healing Assay

Wound healing assays were adapted from the procedure described by Couture et al. [39]. hCECs from the limbal area of a 71-year old donor were used. All cells were grown under 8% CO_2_ at 37 °C and culture medium was changed every 2 days. Human Tissue Engineered Corneas (hTEC) were produced following the self-assembly approach. Briefly, corneal fibroblasts were seeded and cultured in a fibroblast growth medium, and prompted to lay down their own ECM by the addition of 50 µg.mL^−1^ ascorbic acid (Sigma, Oakville, ON, Canada) for 35 days. Reconstructed stromas were produced by the superposition of two tissue sheets. A supplementary week of culture allowed their adhesion to each other. hCECs were then seeded on the surface of the reconstructed stroma and cultured in submerged conditions in complete epithelial cell medium supplemented with ascorbate. After 7 days, reconstructed tissues were fed with an EGF-free epithelial cell medium and raised at the air–liquid interface for 7 days to induce epithelial differentiation. hTECs were then wounded using an 8-mm biopsy punch. A reconstructed stroma made of two more fibroblast sheets was then fixed under the wounded hTECs to allow reepithelialisation to occur. hTECs were fed fresh culture medium every 2 days underneath the fibroblast sheets so that the upper face of the epithelium remained dry. A 50 µL drop of 88 nM of GNP 2000 was added daily to hTECs (directly on the wounded area, on the side facing the air). The specific concentration selected for GNPs was based on the MTS assay results conducted on hCECs. Wound closure was observed macroscopically every 24 h for 5 days and photographed daily with a Zeiss Imager Z2 microscope (Zeiss Canada Ltd., North York, ON, Canada). All experiments were conducted in quadruplicate.

## 4. Conclusions

For different applications, including medical ones, GNPs must be ultrastable in order to increase their efficiency and prevent their aggregation and accumulation in the body. The new experimental conditions based on the use of acetonitrile and presented in the present study led to the synthesis of ultrastable nanoparticles. GNPs with ligands of different molecular weights, ranging from 800 to 6000 g.mol-1, were deeply characterized and found to be ultrastable. No aggregation occurred after several drastic treatments like freeze drying, heating, ultracentrifugation, and autoclaving. Indeed, UV-visible spectroscopy, TEM, and DLS data suggested ultrastability before and after each treatment, highlighting the ability of GNPs to support extreme experimental conditions, some of which are required for ophthalmic drug formulation. The latter properties combined with the results obtained during the drug encapsulation experiments all suggest that these ultrastable GNPs can be potentially used as drug delivery systems. Moreover, ultrastable GNPs showed no signs of cytotoxicity in either MTS or wound healing assays. These characteristics led the authors to consider future experiments on small animals to confirm their ability to be used as slow-releasing drug delivery systems in the treatment of corneal diseases.

## 5. Patents

GNP synthesis is patented (E. Boisselier, V. Pernet, M. Omar, M. Ouellette, Ultrastable gold nanoparticles for drug delivery applications and synthesis thereof, patent WO/2018/102921).

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
