# Peer review of "Synthesis of Ultrastable Gold Nanoparticles as a New Drug Delivery System"

_molecules, 2019, doi:10.3390/molecules24162929_

Round 1

Reviewer 1 Report

Authors provide a methodology for the synthesis of stable gold nanoparticles for their application in drug delivery system.  The stability of the proposed GNPs has been compared with the standard methods Turkevich and Brust method The obtained results are interesting can be improved after addressing the following points:

More detailed schematic representation of all the steps followed for the synthesis and functionalization will help the readers

Three PEG groups with different molecular wt (800, 2000 and 6000) have been used to functionalize the GNPs.  Although the hydrodynamic diameter is found to increase with the increase in the molecular wt., authors need to explain why the core diameter, as well as the number of ligands, is maximum for GNP 2000 (Table 1).

We understand that for the comparison and the effect of mol. wt of PEG, all the three proposed particles are made under the same conditions.  why %S decreases for GNP 2000?

For how many particles core diameter is averaged?

Similar to the table 1 made for the proposed particles, parameters corresponding to the other two kinds of particles (Turkevich and Brust method) should also be included

The impact of the work can be increased by including some details (different steps) and illustrations to explain the mechanism (nucleation, growth, and stability).

Reviewer 2 Report

Masse and colleagues report on a three types of an ultrastable gold nanoparticles stabilized by thiolated PEG. The manuscript considers the important problem related to the gold nanoparticles stability under differed conditions. 

In my opinion, the presented work is very interesting and has been carried out to a high standard. The introduction is well described; the same is with the results and discussion. Several experimental techniques including TEM, DLS, and UV/Vis were applied to control the stability of the obtained nanostructures.

I have only two minor points, which need to be explained by the Authors:

1. Why the addition of acetonitrile was necessery?

2. Why the encapsulation was performed? Please precise more detail. 

Reviewer 3 Report

line 16: The sentence "For comparison purposes, two other conventional nanoparticle types fail to maintain colloidal stability under all conditions." is misleading. I think you wanted to say something like "For comparison purposes, two conventional nanoparticle types were used to assess their colloidal stability under all conditions."?

line 26: "Gold Nanoparticles ..." There is no need to capitalize the first letter of "nanoparticles" here. 

line 35: You criticise the term "ultra-stable" in connection with golf nanoparticles. However, you work with this term in your paper with no precise definition of ultrastability. Are you talking about thermodynamical stability? Is there any need to define "ultra-stability"?

line 41: What "... withstand sterilization by autoclave ..." means? Technically, some parameters of the nanoparticles could be potentially by changed by autoclaving such as particle size. However, "to withstand" is far from science based assessment. Further, some papers suggest that recent gold nanoparticles keep their stability after autoclaving (Yasmin, Nano Corvengence, 2014).

line 118, line 114, line 132 (Fig.1), line 159: Is there any reason why mean and STD of core diameters are presented by a single digit number? You have histograms from TEM images which gives you diameters of some 1000 nanoparticles. E.g. CD 3 +/- 2 nm looks like rough estimation.

line 132 (Fig.1): UV-visible spectra graphs and TEM images are so small that they are almost useless to a reader. I would appreciate an example of GNP 800 UV-visible spectrum graph and TEM image before treatment in full size. Then, the Figure 1 could be converted to a table with numbers describing the differences between control (before treatment) and after treatment. 

line 129: "The UV-visible spectrum for each GNP observed after treatment ... is almost the same as the initial one." Define "almost same". Such assessment is highly insufficient. For example, I can see that UV-visible spectra of GNP 800 before treatment and after autoclaving differ. The figure is small so I can hardly evaluate. The difference must be calculated if you want to assess stability after treatment.

line 312: "... about 300 GNP were counted and analyzed for each sample". I doubt this is correct. Based on histograms in the figure 1, number of GNP counted varied from 150 to 1000.

Round 2

Reviewer 3 Report

I found your answers and changes as satisfactory and treating all my comments and suggestions from the first review.